# Design of Backstepping Control Based on a Softsign Linear–Nonlinear Tracking Differentiator for an Electro-Optical Tracking System

Jiachen Li [1,2,3,4], Shanlin Zhuang [1,2,3,4], Haolin Wang [1,2,3,4], Jiuqiang Deng [1,2,3,4,*] and Yao Mao [1,2,3,4]

1   National Key Laboratory of Optical Field Manipulation Science and Technology, Chinese Academy of Sciences, Chengdu 610209, China; lijiachen22@mails.ucas.ac.cn (J.L.); zhuangshanlin22@mails.ucas.ac.cn (S.Z.); wanghaolin20@mails.ucas.ac.cn (H.W.); maoyao@ioe.ac.cn (Y.M.)
2   Key Laboratory of Optical Engineering, Chinese Academy of Sciences, Chengdu 610209, China
3   Institute of Optics and Electronics, Chinese Academy of Sciences, Chengdu 610209, China
4   University of Chinese Academy of Sciences, Beijing 100049, China
*   Correspondence: jqdeng@ioe.ac.cn

**Abstract:** To address the problems of a low tracking accuracy and slow error convergence in high-order single-input, single-output electro-optical tracking systems, a backstepping control method based on a Softsign linear–nonlinear tracking differentiator is proposed. First, a linear–nonlinear tracking differentiator is designed in conjunction with the Softsign excitation function, using its output as an approximate replacement for the conventional differentiation process. Then, this is combined with backstepping control to eliminate the "explosion of complexity" problem in conventional backstepping procedures due to repeated derivation of virtual control quantities. This reduces the workload of parameter tuning, takes into account the rapidity and stability of signal convergence, and improves the trajectory tracking performance. This method can ensure the boundedness of the system signal. The effectiveness and superiority of this control method are verified through simulations and experiments.

**Keywords:** backstepping control; tracking differentiator; electro-optical tracking system; Softsign function; linear–nonlinear systems

## 1. Introduction

An electro-optical tracking system (ETS) is a piece of complex and high-precision servo tracking equipment integrating optical, mechanical, and electronic technology. It has the characteristics of a fast response speed, a low tracking error, and strong disturbance resistance [1]. These devices are mainly used for real-time tracking and measurements of moving targets, and are widely employed in the fields of target observation, laser communication, quantum communication, aviation, aerospace, automated production, and other fields [2,3]. In the aerospace field, they are used for spacecraft guidance, positioning, attitude control, etc. [4]. In the field of automated production for automated processing, they are used for machine vision [5], target tracking and pointing, motion target trajectory measurements [6], etc. In the medical field, they are used for target tracking in surgical robots, etc.

For this type of system, some scholars have proposed PID control, sliding mode control, adaptive control, and neural network control methods, among others. These control methods have effectively improved the tracking performance of ETS control from different aspects. PID control is a simple and effective control method that meets the steady-state and dynamic performance requirements of a system by adjusting three parameters: the proportional, the integral, and the derivative. However, for electro-optical tracking systems with nonlinearity and uncertainty, the performance of PID control may be limited [7].

Sliding mode control uses a sliding surface and switching logic to achieve the desired system performance, exhibiting strong robustness and adaptability. However, it may suffer from chattering issues [8]. Fuzzy control is a control method based on fuzzy logic, capable of handling uncertainty and nonlinear problems. Yet, in high-dimensional and complex electro-optical tracking systems, the design of fuzzy rules may become complex and time-consuming [9]. Neural network control is an artificial-intelligence-based control method suitable for highly nonlinear and uncertain systems. However, training neural networks requires substantial amounts of data and time, and there may be some delay in real-time applications [10].

Furthermore, an electro-optical tracking system is limited by the sensor frequency and the performance of the driving mechanism, which leads to the problems of a low tracking accuracy and slow error convergence. To address this problem, backstepping control techniques [11–13] have been developed rapidly in recent years. Backstepping control is not only characterized by an ease of design and implementation but also by the ability to measure the state of the system in real time and adjust the inputs according to the difference between the target output and the actual output, thus achieving highly accurate tracking. It can also effectively suppress the instability of the system and thus ensure its stability. A backstepping design allows the control scalar functions and controllers to be systematic, structured, and jitter-free, making them widely available [14,15].

The backstepping control method suffers from the "complexity explosion" problem [16,17], which leads to a complex solution process and slow convergence.

In the process of backstepping control design, the quality of the position feedback signal, the speed feedback signal, etc., is key to improving the control accuracy, and it is largely influenced by signal filtering and signal differentiation. According to the position signal, the traditional differentiation method can calculate a series of differential signals such as the velocity and acceleration. There are multiple successive differentials in backstepping control, which can easily lead to the problem of a "complexity explosion" [16,17]. The traditional differentiation process can reduce the accuracy of the signal differentiation estimation when it is affected by signal noise, and, in turn, the control effect of the backstepping controller is affected.

To avoid a "complexity explosion" and to suppress signal noise, past studies usually used filters and various signal processing methods to preprocess the signal, such as dynamic surface control (DSC) [18–21] and command filtering (CF) control [22–24]. The main feature of DSC is the introduction of a first-order filter in the backstepping design process to replace the differential operator in each virtual controller design step; however, the DSC scheme does not consider the impact of filtering errors on the control system [25]. To solve this problem, researchers proposed a command-filtered [23,24,26] backstepping control method by introducing virtual input second-order filtering in each step of the conventional backstepping design process to replace the conventional differential process and by using an error compensation mechanism to overcome the shortcomings of DSC. However, the transient performance of the filtered signal of the conventional command filter in this scheme is poor and the performance of its differential process decreases when the frequency of the input signal increases.

To solve these problems, different tracking differentiators have been used to improve the transient performance of the filtered signal and the differentiation performance at higher frequencies [27–34]. These differentiators can complete the process of tracking and differentiating a real-time signal without relying on the controlled object model, and they use an integration process instead of the differentiation process used in the traditional numerical differentiation method [35] to avoid a "complexity explosion" while performing signal filtering. With in-depth research on tracking differentiator technology in recent years, various types of improved tracking differentiators, such as Linear Tracking Differentiators (LTDs) [27], High-Speed Tracking Differentiators (HSTDs) [28], a New Simple Linear Tracking Differentiator (T-D) [29], and High-Gain Tracking Differentiators (HGTDs) [30], have been gradually developed to improve the dynamic performance of differentiators.

However, the tracking function of these tracking differentiators is not able to simultaneously consider the rapidity and stability of signal convergence, which may cause the rapidity or stability of signal convergence to deteriorate when adjusting the convergence trend near the equilibrium point.

When a tracking differentiator has a relatively fast convergence speed, the speed of the change in state quantities near the equilibrium point increases simultaneously, which leads to convergence chattering and other problems, resulting in a decrease in the tracking accuracy of the differentiator. In this regard, Arctangent Tracking Differentiators (ATDs) in the form of an inverse tangent using an inverse tangent function [31], Modified Tracking Differentiators (MTDs) designed using a nonlinear odd-exponential continuous function that is stable at only one equilibrium point [32], New Nonlinear–Linear Tracking Differentiators (NTDs) using hyperbolic tangent (Tanh) functions [33], and Hyperbolic-Sine-Based Tracking Differentiators (HNTDs) [34] using hyperbolic sinusoidal functions have been proposed, which are based on a common improvement strategy: introducing both nonlinear and linear links into the differentiator design. Linear and nonlinear links exhibit different degrees of action when the state is far from or close to the equilibrium point, ensuring the rapidity and stability of differentiator convergence. However, the structural form of a tracking differentiator designed in this way is relatively complex, with more parameters, and the functions have the problem of a faster gradient disappearance, and so a more accurate approximate differentiation process may not be achieved.

In summary, in high-order electro-optical tracking systems characterized by nonlinearity and uncertainty, the performance of PID control may be limited [7], sliding mode control may encounter significant chattering issues [8], fuzzy control requires the design of complex fuzzy rules [9], and neural network control demands substantial additional data and time for training [10]. Therefore, the decision has been made to adopt backstepping control, which can circumvent these issues and provide effective control for an electro-optical tracking system. When solving the complexity explosion present in backstepping control, the existing dynamic surface will introduce filtering errors and reduce the tracking accuracy [21]; the transient performance of the filtered signal of the existing command filter is poor [24]; and existing tracking differentiators have a complex structure and more parameters and the gradient of the function they use disappears faster, so it is not possible to realise an approximate differentiation process with the proposed accuracy [34]. Thus, this paper proposes a linear–nonlinear tracking differentiator based on the Softsign excitation function (SL-NTD) for an electro-optical tracking system. The results show that the proposed control design ensures that all signals are in a bounded set and the tracking error converges to the desired neighborhood of the origin. Compared to all existing technology, the innovations and main contributions of the proposed control scheme can be summarized as follows:

1.  A linear–nonlinear tracking differentiator is designed using the Softsign excitation function for the first time. Compared to the dynamic surface used in Liang and Qiu's work, the method proposed in this paper does not introduce filtering errors [21]; compared to the command filter used in Han and Yu's work, the method in this paper can simultaneously take into account the speed and stability of signal convergence [24]; and compared to the tracking differentiator used in Fan and Jing's work, the method in this paper has fewer parameters, the disappearance of the gradient is slowed down, and thus differentiation can be approximated more accurately [34].
2.  This paper proposes introducing a Softsign tracking differentiator in each step of backstepping control for the first time, using an approximation of the output of the tracking differentiator instead of the traditional differentiation process in virtual control, which solves the problem of the "complexity explosion" in backstepping control.
3.  The number of parameters in the whole backstepping control process is significantly reduced, improving parameter tuning in the scheme. Finally, the feasibility and superiority of the backstepping control method of the linear–nonlinear tracking differentiator based on the Softsign excitation function are verified via simulations and experiments.

The remainder of the paper is structured as follows. Section 2 presents the problem description and introduces the research objectives and methodology of the paper. Section 3 describes the backstepping control design process and the proof of its stability for a linear–nonlinear tracking differentiator based on the Softsign excitation function. Section 4 presents a specific control object example and compares the proposed method with other methods in simulations and experiments to solve the "complexity explosion" problem. The effectiveness and superiority of the proposed design method are verified in simulations and experiments. Section 5 provides the conclusions of the paper.

## 2. Problem Description

Consider the following electro-optical tracking system that can be expressed as:

$$\begin{cases} \dot{x}_1 = x_2 \\ \dot{x}_2 = x_3 \\ \cdots\cdots \\ \dot{x}_{n-1} = x_n \\ \dot{x}_n = u \\ y = x_1 \end{cases} \tag{1}$$

In Equation (1), $x = [x_1, x_2, \ldots, x_n]^T \in R^n$ is the state quantity, and it is measurable, $u$ is the control rate, and $y$ is the output.

The ultimate goal of this paper is to make the class of ETSs achieve high-accuracy trajectory tracking, where the root mean square error can be used as a reference for adjustment and measurements, as shown in Equation (2) below:

$$RMSE = \sqrt{\frac{1}{N}\sum(v_t - x_1)^2} \tag{2}$$

where $v_t$ is the input standard trajectory and $x_1$ is the output actual trajectory. In order to improve the trajectory tracking accuracy, it is necessary to reduce the error between the actual trajectory and the standard trajectory; therefore, in this paper, a new backstepping controller is designed that achieves high-precision tracking while ensuring system stability by measuring the system state in real time, adjusting the input according to the difference between the target output and the actual output, and using the state feedback information of the system to offset the effects of perturbation.

The backstepping control method used in this paper is a commonly used control method [11,12], the basic idea of which is decomposing a complex system into subsystems that do not exceed the order of the system. As shown in Equation (1) above, the system can be divided into n subsystems, and then a Lyapunov function $V_n$ and an intermediate virtual control quantity $x_{(i+1)d}$ can be designed, respectively, for each subsystem $x_{i+1}$. In each subsystem, $\dot{x}_i = x_{i+1}$ is virtual control, and appropriate virtual feedback $x_{i+1} = x_{(i+1)d}$, where $i = 1, 2, \ldots, n-1$, allows the previous state of the system to reach asymptotic stability and then "back up" the entire system until the entire controller design is completed. In this process, it is necessary to differentiate the virtual control quantity $x_{(i+1)d}$ used in each step, and after simplification, this is equivalent to multiple successive differentiations of the first virtual control quantity $x_{1d}$. This can lead to the phenomenon of a "complexity explosion". To solve the problem of the "complexity explosion ", this paper uses a Softsign tracking differentiator to replace the traditional differential derivation process.

## 3. SL-NTD-Based Backstepping Controller Design

### 3.1. Description of the Softsign Tracking Differentiator

In this paper, we propose a linear–nonlinear tracking differentiator based on the Softsign excitation function. This section focuses on the introduction of the Softsign excitation function; an image of the function is shown in Figure 1, and the mathematical equation is of the form shown in Equation (3):

$$Softsign(x) = \frac{x}{1 + |x|} \tag{3}$$

where $x$ is the independent variable of the Softsign excitation function, and the value of $Softsign(x)$ converges to 1 when the value of $x$ is positive infinity and to $-1$ when the value of $x$ is negative infinity.

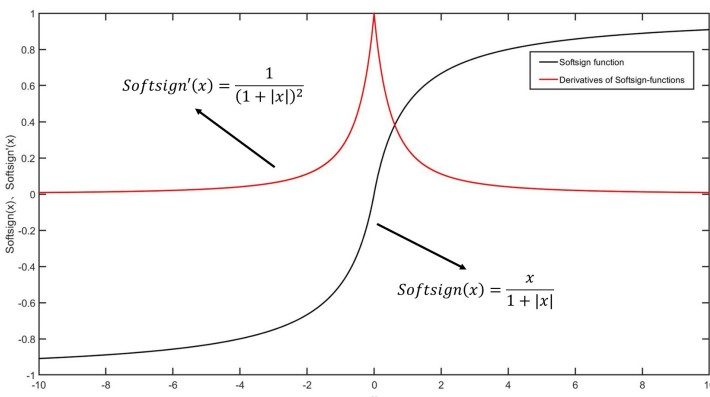

**Figure 1.** Softsign function image.

The input value of the Softsign function is small and its output value in the interval close to 0 presents non-linear characteristics, so it can eliminate or alleviate the problem of jittering to a certain extent.

The magnitude of the Softsign function and the coefficients of the independent variables can be extended to obtain the function $f = m * Softsign(nx)$, and changing the values of $m$ and $n$ can change the magnitude of the Softsign function and the rate of change. Changing the magnitude of $m$ changes the range of function values; when $n$ decreases to nearly 0, the output of the function will tend to be linear and the slope will decrease as well. This indicates that as $n$ decreases, the function $f$ becomes more linear, smoother, and more saturated. This characteristic can alleviate or even eliminate the impact of jitter and can also reduce the risk of overfitting in many cases, making the output more stable.

In addition, it can be seen from Figure 1 that the Softsign function has properties such as smoothness and continuity, and the saturation interval is small, giving it a slower decreasing derivative, which also indicates that it can learn more efficiently and can better solve the problem of gradient disappearance. Thus, it is suitable and convenient for use in the design of tracking differentiators.

In order to design a linear–nonlinear tracking differentiator based on the Softsign excitation function, the following theorem/lemma and related proofs need to be given first.

**Theorem 1.** *There is the following system:*

$$\begin{cases} \dot{k}_1(t) = k_2(t) \\ \dot{k}_2(t) = -a * [k_1(t) + Softsign(k_1(t))] - b * [k_2(t) + Softsign(k_2(t))] \end{cases} \tag{4}$$

*If the parameters a and b in the above system are greater than 0, then the system is uniformly asymptotically stable at the origin $(0,0)$. This means that for any initial condition, the system state will converge to the origin as time tends to infinity and the rate of convergence is consistent for all initial conditions.*

**Remark 1.** *When there exist systems of the form described above, whose parameters are all greater than 0, then the system is uniformly asymptotically stable at the origin $(0,0)$.*

**Proof.** First, construct the Lyapunov function:

$$V(k_1, k_2) = \int_0^{z_1} a * Softsign(\theta) d\theta + \frac{a}{2} * K_1^2 + \frac{1}{2} * K_2^2 \tag{5}$$

Theorem 1 gives $a > 0$ and $a * Softsign(k_1) > 0$ when $k_1 > 0$, and $a * Softsign(k_1) < 0$ when $k_1 < 0$. Then, from the median integral theorem, we have

$$\int_0^{z_1} a * Softsign(\theta) d\theta = a * Softsign(\sigma) * k_1 > 0 \tag{6}$$

where $0 < \sigma < k_1$; thus, it is obtained that $\int_0^{z_1} a * Softsign(\theta) d\theta > 0$.

When $k_1 \neq 0$ and $k_2 \neq 0$, then $\frac{a}{2} * k_1^2 > 0$ and $\frac{1}{2} * k_2^2 > 0$; therefore, we obtain $V(k_1, k_2) > 0$. Take the derivative of the Lyapunov function:

$$\begin{aligned}
\dot{V}(k_1, k_2) &= (a * Softsign(k_1) + a * k_1) * \dot{k}_1 + k_2 * \dot{k}_2 \\
&= a * k_2 * Softsign(k_1) + a * k_1 * k_2 + k_2 * [-a* \\
&\quad (k_1 + Softsign(k_1)) - b * (k_2 + Softsign(k-2))] \\
&= -b * k_2^2 - b * k_2 * Softsign(k_2)
\end{aligned} \tag{7}$$

Since Theorem 1 gives $b > 0$, it can be obtained that $b * k_2 * Softsign(k_2) > 0$ and $\dot{V}(k_1, k_2) \leq 0$. It follows that $\dot{V}(k_1, k_2) = 0$ if and only if $k_2 = 0$ near (0.0). Therefore, according to Lyapunov's second theorem, the system of Theorem 1 is asymptotically stable and so Theorem 1 is valid. $\square$

**Lemma 1.** *Due to the nature of the Softsign function, the system can be divided into two action phases when it comes to the specific role:*

1. *When $|k| > 2.5$, the system is far away from the equilibrium position, $|k| > |Softsign(k)|$. This has a major role in driving system (4), described by:*

$$\begin{cases} \dot{k}_1(t) = k_2(t) \\ \dot{k}_2(t) = -a * k_1(t) - b * k_2(t) \end{cases} \tag{8}$$

2. *When $|k| < 2.5$, the system is far away from the equilibrium position, $|k| < |Softsign(k)|$. This has a major role in driving system (4), described by:*

$$\begin{cases} \dot{k}_1(t) = k_2(t) \\ \dot{k}_2(t) = -a * Softsign(k_1(t)) - b * Softsign(k_2(t)) \end{cases} \tag{9}$$

**Remark 2.** *The tracking differentiator constructed from the Softsign excitation function can be divided into two action phases in the specific role, which are also each asymptotically stable and allow the system state to always converge quickly and steadily. The change in state is shown in Figure 2, where the origin (0,0) is the equilibrium point.*

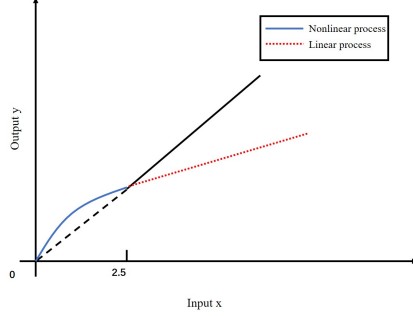

**Figure 2.** Status changes in the two phases of action of the tracking differentiator.

**Proof.** From Theorem 1 and Theorem 2 in [28] it is easy to prove that Equations (8) and (9) are also asymptotically stable and enable the fast and stable convergence of the system state.

Therefore, a nonlinear–linear tracking differentiator based on the Softsign excitation function, referred to as SN-LTD, is designed. The control block diagram is shown in Figure 3, and the mathematical model is shown in Equation (10). □

**Theorem 2.** *For the following system:*

$$\begin{cases} \dot{k}_1(t) = k_2(t) \\ \dot{k}_2(t) = (a * ((x_1(t) - v(t)) + Softsign(x_1(t) - v(t))) \\ + b * (\frac{x_2(t)}{R} + Softsign(\frac{x_2(t)}{R}))) * (-R^2) \end{cases} \tag{10}$$

*where $v(t)$ is the input signal, $x_1(t)$ and $x_2(t)$ are variables, and $x_1(t)$ is the input signal after filtering, $x_2(t)$ is the extracted differential signal, and a and b are parameters greater than 0. The solution of this tracking differentiator system satisfies $\lim_{x \to \infty} \int_0^T |x_1(t) - v(t)dt = 0|$ at $T > 0$.*

According to Theorem 1 and Lemma 1, it is easy to prove that Theorem 2 holds.

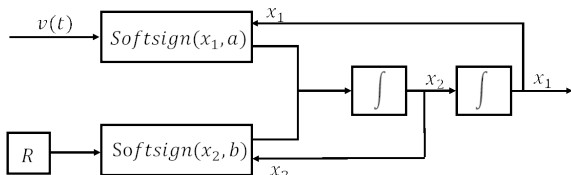

**Figure 3.** Control block diagram based on the Softsign linear–nonlinear tracking differentiator.

Through the derivation of the formula for the Softsign-based tracking differentiator, it can be found that the output of the Softsign function tends to be 1 when the difference between $x_1(t)$ and $v(t)$ is large, and at this time, the output of the Softsign-based tracking differentiator mainly depends on the difference between $x_1(t)$ and $v(t)$. The method can effectively track the changes in the system. At the same time, the Softsign-based tracking differentiator can better overcome the differential explosion problem because the Softsign function has the ability to effectively filter out noise in edge detection. Also, the range of output signals of the Softsign-based tracking differentiator is wide, effectively avoiding the output saturation problem. This means that the Softsign-based tracking differentiator has better adaptability in dealing with nonlinear systems.

**Theorem 3.** *The theoretical analysis of the Softsign-based tracking differentiator is compared with an ordinary linear tracking differentiator, and the results show that the Softsign-based tracking differentiator has a faster convergence speed and smaller errors. The theoretical analysis process is as follows:*

*The state space equation for the linear tracking differentiator used for the comparison is:*

$$\begin{cases} \dot{x}_1(t) = x_2(t) \\ \dot{x}_2(t) = (-a_0 * (x_1(t) - v(t)) - a_1 * \varepsilon * x_2(t)) * \varepsilon^{-2} \end{cases} \tag{11}$$

*where $v(t)$ is the input signal, $x_1(t)$ and $x_2(t)$ are the variables, and $a_0$, $a_1$, and $\varepsilon$ are parameters greater than 0, where $a_1 = 2$, $a_0 = 1$, $\varepsilon = 0.004$, and $a = 7$ and $b = 8$ in Equation (4).*

Define the Lyapunov function as $V = x_1^2 + v_2^2$, and for a linear tracking differentiator, calculate the derivative of the Lyapunov function as:

$$\begin{aligned} \dot{V}_1 &= 2 * x_1 * \dot{x}_1 + 2 * x_2 * \dot{x}_2 \\ &= 2 * x_1 * x_2 + 2 * x_2 * (-2 * (x_1 - v) - 0.004 * x_2)/0.004^2 \\ &= 2 * x_1 * x_2 + 2 * x_2 * (-125000 * (x_1 - v) - 250 * x_2) \end{aligned} \tag{12}$$

For the Softsign-based tracking differentiator, the derivative of the Lyapunov function is calculated as:

$$
\begin{aligned}
\dot{V}_1 =& 2 * x_1 * \dot{x}_1 + 2 * x_2 * \dot{x}_2 \\
=& 2 * x_1 * x_2 + 2 * x_2 * (-2520000 * (x_1 - v) - 3000 * x_2 - 2520000 * softsign(x_1 - v) \\
& - 1800000 * softsign(x_2/600))
\end{aligned}
\tag{13}
$$

A comparison shows that $\dot{V}_1 > \dot{V}_2$ and the Softsign-based tracking differentiator has faster convergence and more minor errors according to the Lyapunov stability theorem. Similarly, it can be proven that the Softsign-based tracking differentiator has more advantages over other differentiators.

**Remark 3.** *Compared to the command filters mentioned in the literature [23,24], the use of a Softsign-based tracking differentiator, which uses the input $v(t)$ to obtain $x_1(t)$ and $\dot{x}_1(t)$, improves not only the transient performance of the filtered signal but also the differentiation performance at higher frequencies.*

**Remark 4.** *Compared with various types of tracking differentiators mentioned in the literature [27–34], the use of the Softsign-based tracking differentiator, which uses the input $v(t)$ to obtain $x_1(t)$ and $\dot{x}_1(t)$, simplifies the structure of the tracking differentiator and reduces the number of parameters while taking into account the rapidity and stability of signal convergence, better solving the "complexity explosion" problem.*

*3.2. Design of Backstepping Control Based on the Softsign Tracking Differentiator*

Consider the following nth-order single-input and single-output electro-optical tracking system:

$$
\begin{cases}
\dot{x}_1 = x_2 + f_1(x_1) \\
\dot{x}_2 = x_3 + f_2(x1, x2) \\
\cdots\cdots \\
\dot{x}_{n-1} = x_n + f_{n-1}(x_1, x_2, \ldots, x_{n-1}) \\
\dot{x}_n = u + f_n(x_1, x_2, \ldots, x_n)
\end{cases}
\tag{14}
$$

where $x_1, x_2, \ldots, x_n$ is the state quantity, $u$ is the control rate, the input is $x_{1d}$, and the output is $x_1$. In addition, $x_{2d}, x_{3d}, \ldots, x_{nd}$ is the virtual input quantity and also the expected value of the state quantity.

Design the control rate according to the conventional backstepping control design idea. In each $x_{i+1}$ subsystem, $\dot{x}_i = x_{i+1} + f_i(x_1, x_2, \ldots x_i)$ is virtual control, and appropriate virtual feedback $x_{i+1} = x_{(i+1)d}$, where $i = 1, 2, \ldots, n-1$, makes the previous state of the system reach asymptotic stability, but the solution of the system generally does not satisfy $x_{i+1} = x_{(i+1)d}$. For this reason, error variables are introduced in the hope that some asymptotic properties between $x_{i+1}$ and the virtual feedback $x_{(i+1)d}$ can be achieved through the action of control, and thus the asymptotic stability of the whole system can be achieved.

The derivative $\dot{x}_{1d}, \dot{x}_{2d}, \ldots, \dot{x}_{nd}$ of the virtual control quantity $x_{1d}, x_{2d}, \ldots, x_{nd}$ designed in the following is substituted by the differential signal $\hat{\dot{x}}_{1d}, \hat{\dot{x}}_{2d}, \ldots, \hat{\dot{x}}_{nd}$ extracted by a linear–nonlinear tracking differentiator based on the Softsign excitation function which is provided to the backstepping controller.

Assume that the n tracking errors are: $z_1, z_2, \ldots, z_n$

$$
\begin{aligned}
z_1 &= x_1 - x_{1d} \\
z_2 &= x_2 - x_{2d} \\
&\cdots\cdots \\
z_n &= x_n - x_{nd}
\end{aligned}
\tag{15}
$$

Dividing the nth-order system into n first-order subsystems, n Lyapunov functions are defined in turn:

$$V_1 = \frac{1}{2} * z_1^2 \tag{16}$$

$$V_2 = \frac{1}{2} * z_1^2 + \frac{1}{2} * z_2^2 \tag{17}$$

$$\ldots\ldots$$

$$V_n = \frac{1}{2} * z_{n-1}^2 + \frac{1}{2} * z_n^2 \tag{18}$$

Then, design the backstepping control process from top to bottom in turn. The design process needs to ensure that $\dot{V}_n \leq 0$, which ensures system stability.

Differentiate the n Lyapunov functions in turn, as follows:

$$\dot{V}_1 = z_1 * \dot{z}_1 = z_1 * (\dot{x}_1 - \dot{\tilde{x}}_{1d}) \tag{19}$$

$$\dot{V}_2 = -k_1 * z_1^2 + z_2 * \dot{z}_2 = -k_1 * z_1^2 + z_2 * (\dot{x}_2 - \dot{\tilde{x}}_{2d}) \tag{20}$$

$$\ldots\ldots$$

$$\dot{V}_n = -k_{n-1} * z_{n-1}^2 + z_n * (z_{n-1} + \dot{z}_n) \tag{21}$$

**Subsystem 1:** In order for the system to be stable, $\dot{V}_1 \leq 0$, i.e., negative definite. At this point, the above requirement can be satisfied as long as $\dot{x}_1 - \dot{x}_{1d}$ tends to $-k_1 * z_1$, $(k_1 > 0)$, so that the following is obtained:

$$\dot{V}_1 = -k_1 * z_1^2 \tag{22}$$

At this point, set $x_{2d}$ as the tracking object of $x_2$, and because the system itself $\dot{x}_1 = x_2 + f_1(x_1)$, we can get:

$$x_{2d} = \dot{\tilde{x}}_{1d} - k_1 * z_1 - f_1(x_1) \tag{23}$$

**Subsystem 2:** In order for the system to be stable, $\dot{V}_2 \leq 0$, i.e., negative definite. At this point, the above requirement can be satisfied as long as $\dot{x}_2 - \dot{x}_{2d}$ converges to $-k_2 * z_2$, $(k_2 > 0)$, so that the following is obtained:

$$\dot{V}_2 = -k_1 * z_1^2 - k_2 * z_2^2 \tag{24}$$

At this point, set $x_{3d}$ as the tracking object of $x_3$, and because the system itself $\dot{x}_2 = x_3 + f_2(x_1, x_2)$, we can get:

$$x_{3d} = \dot{\tilde{x}}_{2d} - k_2 * z_2 - f_2(x_1, x_2) \tag{25}$$

......

**Subsystem n:** Here, to ensure system stability, it is necessary that $\dot{V}_n < 0$, i.e., negative definite.

As $z_{n-1} + \dot{z}_n = -k_n * z_n$, where $k_n > 0$, we can obtain:

$$\dot{V}_n = -k_{n-1} * z_{n-1}^2 - k_n * z_n^2 \tag{26}$$

Solving the equation $z_{n-1} + \dot{z}_n = -k_n * z_n$ yields:

$$z_{n-1} + (\dot{x}_n - \dot{\tilde{x}}_{nd}) = -k_n * z_n \tag{27}$$

$$z_{n-1} + u + f_n(x_1, x_2, \ldots, x_n) - \dot{\tilde{x}}_{nd} = -k_n * z_n \tag{28}$$

The final control rate is then obtained:

$$u = \dot{\tilde{x}}_{nd} - k_n * z_n - z_{n-1} - f_n(x_1, x_2, \ldots, x_n) \tag{29}$$

Because the errors are exponentially asymptotically stable, and thus so is the control rate designed above, the original nonlinear system is guaranteed to be exponentially asymptotically stable.

To sum up, the design block diagram of the proposed control strategy is summarized in Figure 4.

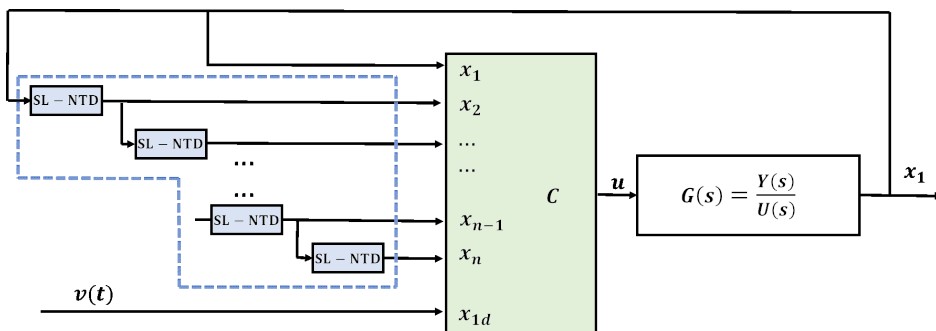

**Figure 4.** Backstepping control block diagram based on the Softsign linear–nonlinear tracking differentiator.

**Remark 5.** *The modified tracking differentiator SL-NTD in Figure 4 is used to generate alternative differential signals. In fact, to facilitate the implementation process and reduce the number of control parameters in the developed control strategy, all SL-NTDs can implemented with the same structure.*

**Remark 6.** *The convenience of the Softsign tracking differentiator-based backstepping control method proposed in this paper is that the traditional backstepping design process can still be used, and when the differentiation step is encountered, the differentiation extracted by the Softsign tracking differentiator is used instead, without breaking the traditional design steps. The tracking differentiator and backstepping control are independent of each other and can be used separately, which is more convenient.*

### 3.3. Proof of Stability

**Lemma 2.** *The second method of Lyapunov (direct method).*

Stability theorem: For a continuous nonlinear system, if one can construct a scalar function $V(x)$ with continuous first-order partial derivatives with respect to $x$, $V(0) = 0$ and an attraction region $\Omega$ is present around the origin of the state space such that all nonzero states $x \in \Omega$ satisfy the following conditions.

1.  $V(x) > 0$.
2.  $\dot{V}(x) \leq 0$.
3.  For any nonzero $x \in \Omega$, $\dot{V}(\varnothing(t; x_0, 0)) \not\equiv 0$.

Then, the original system equilibrium state $x = 0$ is asymptotically stable in the $\Omega$ region.

**Proof.** The nth-order system is divided into n first-order systems using n Lyapunov functions to maintain stability, as shown in Equations (16)–(18), where there are n-3 more equations between Equations (17) and (18). Looking at Equations (16)–(18), we can see that $V_1, V_2, \ldots, V_n$ are all constant and greater than or equal to zero.

Deriving the above n equations separately, after introducing the virtual control quantity, we obtain Equations (22), (24) and (26), where there are n-3 equations between Equations (24) and (26). Observe that in Equation (22), since $k_1$ is defined to be a constant greater than zero, it is guaranteed that $\dot{V}_1$ is constantly less than or equal to zero. Observe that in Equation (24), since $k_1$ and $k_2$ are defined as constants greater than zero, it is guaranteed that $\dot{V}_2$ is constantly less than or equal to zero. Similarly, the n-3 equations between Equations (24) and (26) also prove that $\dot{V}_3, \dot{V}_4, \ldots, \dot{V}_{n-1}$ are constantly less than or equal to zero. Substituting the control rate obtained from Equation (29) into the solved Equation (18), we can obtain Equation (26), since $k_{n-1}$ and $k_n$ are defined as constants greater than zero, so we can guarantee that $\dot{V}_n$ is constantly less than or equal to zero.

Summing up the above conclusions, the following equations can be obtained:

$$V_1 = \frac{1}{2} * z_1^2 \geq 0 \tag{30}$$

$$V_2 = \frac{1}{2} * z_1^2 + \frac{1}{2} * z_2^2 \geq 0 \tag{31}$$

$$\cdots\cdots$$

$$V_n = \frac{1}{2} * z_{n-1}^2 + \frac{1}{2} * z_n^2 \geq 0 \tag{32}$$

$$\dot{V}_1 = -k_1 * z_1^2 \leq 0 \tag{33}$$

$$\dot{V}_2 = -k_1 * z_1^2 - k_2 * z_2^2 \leq 0 \tag{34}$$

$$\cdots\cdots$$

$$\dot{V}_n = -k_{n-1} * z_{n-1}^2 - k_n * z_n^2 \leq 0 \tag{35}$$

According to Lyapunov's stability theory, the controller is asymptotically stable at the origin, which ensures the stability of the whole system. Lemma 2 is proven. □

*3.4. Parameter Selection Guidelines*

The parameters involved in the design of the backstepping control of the optoelectronic tracking system based on the Softsign linear–nonlinear tracking differentiator include the parameters $a$ and $b$ of the improved tracking differentiator and the parameters $k_1$, $k_2$ and $k_3$ of the backstepping control. These parameters need to be considered in the design of the backstepping control to improve the tracking performance of SL-NTDBSC so that the differential signal generated by the Softsign linear–nonlinear tracking differentiator will not cause the "complexity explosion" problem during the whole process and meet the stability requirements. The differential signals generated by the Softsign linear–nonlinear tracking differentiator should be considered in the backstepping control design so that the whole backstepping process will not have the problem of a "complexity explosion" and satisfy the stability requirements. To make the differential signal generated by Softsign linear–nonlinear tracking differentiator closer to the real differential signal and ensure that is does not have the "complexity explosion" problem after multiple differentiations, the tracking differentiator should be designed to combine the fast signal convergence and stability. The parameter tuning guidelines are summarized as follows.

1.  Firstly, when designing SL-NTDBSC, it is necessary to ensure that the adjusted parameters can ensure the stability of the system. According to Theorem 2 and Lemma 2, it can be obtained that the system is stable on the premise of $k_1 > 0$, $k_2 > 0$, $k_3 > 0$, $a > 0$ and $b > 0$.
2.  Then, to meet the requirements related to the response speed and overshoot, the following must be considered. In the Softsign linear–nonlinear tracking differentiator, adjusting $a$ and $b$ can affect the convergence speed of the system, adjusting $a$ too much will result in a large overshoot, increasing $b$ can speed up the response time of the system, and a too small value of $b$ will reduce the convergence speed of the system, and vice versa will lead to system instability. Increasing $k_1$ can speed up the system's response in backstepping control, but a too large $k_1$ may lead to system instability. Similarly, adjusting $k_2$ and $k_3$ can affect the system's overshoot, and suitable values of $k_2$ and $k_3$ can reduce the amount of overshoot.
3.  Finally, the parameters involved above are adjusted appropriately to optimize their quality, and the control parameters are configured as $a = 7$, $b = 4$, $k_1 = 100$, $k_2 = 30$ and $k_3 = 15$ (verified in the experiments in Section 4). Note that the parameter selection in this method is based on an empirical approach.

## 4. Simulation and Experimental Verification

### 4.1. Description of the Backstepping Design of the Specific Controlled Object

In this section, the performance of the backstepping controller based on the improved Softsign tracking differentiator is validated using a precision tracking platform, as shown in Figure 5. The stabilized tracking platform consists of a laser, a charge-coupled device (CCD), a motor, a static mirror, a control module, and other accessories. In this experimental setup, the laser light source is a 635 nanometer fiber-coupled laser diode produced by Thorlabs. The motor used is a voice coil motor with the model number BEI KIMCO LA12-17-000A, and the CCD model is TMC-6740CL, manufactured by NI Corporation based in Austin, TX, USA. The reflective mirror is a custom-designed component. Additionally, the control board employed is the PC104LX3073 from Shenzhen Shenlanyu Technology Co., Ltd. (Shenzhen, China), equipped with the real-time operating system VxWorks. Throughout the experiment, all sampling sensors were operated at a sampling rate of 5000 Hz.

Firstly, driven by the voice coil motor, the target light signal is reflected in the CCD by the tilted mirror, and the target off-target quantity is obtained through filtering and decoding the signal processing module, which is externally fed back to the controller. Secondly, the position, speed and acceleration information of the optical platform controlling the light beam is measured by the sensors mounted on the precision tracking platform in real time, which is internally fed back to the controller. The controller receives the feedback quantity and outputs a control signal to the driver. Finally, after receiving the feedback, the controller outputs a control signal to the driver, which controls the motor to drive the optical platform so that the tracking target is in the center of the CCD target surface to achieve the purpose of precise tracking [36,37].

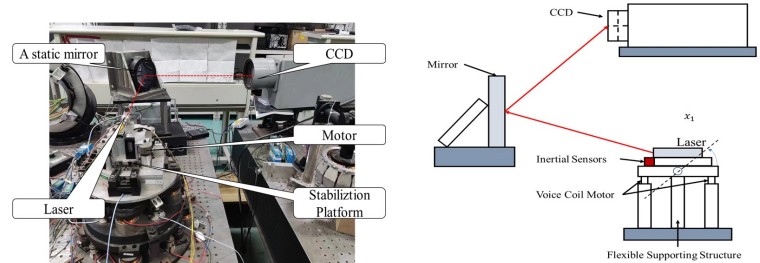

(a) Physical Representation of the Precision Tracking Platform    (b) Schematic Diagram of the Precision Tracking Platform

**Figure 5.** Precision tracking platform.

The system shown in Figure 5 is nonlinear, and its nonlinear characteristics are given as shown in Figure 6. Backstepping control can handle the nonlinear system well. Firstly, the nonlinear part is uniformly classified as a nonlinear term, then the nonlinear part is estimated by designing a new Lyapunov function. Finally, it is compensated for so that the system is unaffected by the nonlinear part. The nonlinear part is estimated to be $\gamma z_3$, where $z_3$ is the third sub-system error during the backstepping design process and $\gamma$ is a gain that is greater than zero. To simplify the expression, in this work, the nonlinear system is linearized directly without considering the nonlinear part.

By inputting the frequency sweep signal into the precision tracking platform, the position-controlled object is measured as shown in Figure 7, where the blue solid line is the actual measured curve and the red dashed line is the fitted curve. After executing the object recognition technique, the expression of the position-controlled object in the frequency domain is derived as:

$$G(s) = \frac{Y(s)}{U(s)}$$
$$= \frac{13390}{0.1508 * s^3 + 40.73 * s^2 + 973.2 * s + 53580} \tag{36}$$

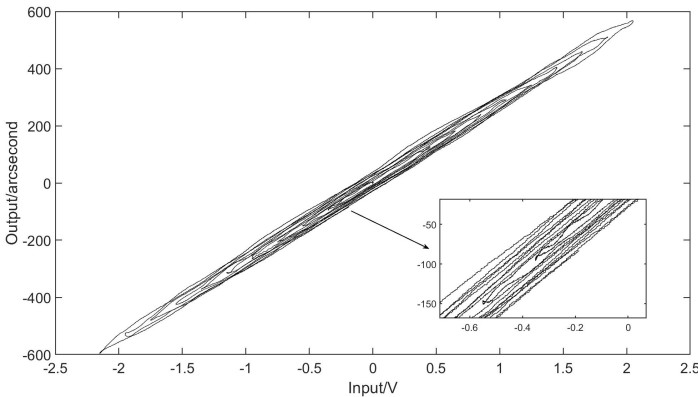

**Figure 6.** Experimental research on system nonlinearity.

The method proposed in this paper will be compared with the command filter, existing tracking differentiators, etc., using the following specific controlled objects.

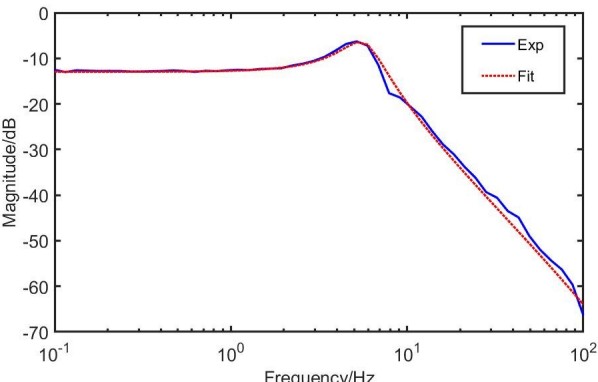

**Figure 7.** Controlled objects of the precision tracking platform.

By dividing the third-order system into three first-order systems and designing the Lyapunov functions in turn, the control rate is finally obtained according to the backstepping design rule as:

$$
\begin{aligned}
u = & \frac{53580 * x_1 + 973.2 * x_2 + 40.73 * x_3}{13390} \\
& + \frac{0.1508 * (\dot{\tilde{x}}_3 d - k_3 * z_3 - z_2)}{13390}
\end{aligned}
\tag{37}
$$

where, in the Softsign improved tracking differentiator-based backstepping controller, $\dot{\tilde{x}}_3 d$ denotes the approximate differential signal of $x_3$ extracted by the Softsign improved tracking differentiator.

### 4.2. Simulation Analysis

Firstly, the differentiation effect of the Softsign linear–nonlinear tracking differentiator designed in this paper is verified by MATLAB/Simulink simulations.

The differentiation effect of the Softsign-based linear–nonlinear tracking differentiator in this paper is first simulated and compared with the differentiation effect of command filtering. The three plots in Figure 8a–c are given for a comparison of the differentiation trajectories at the input signal frequencies of $1/(2*pi)$ $10/(2*pi)$, and $20/(2*pi)$, respectively, and it can be seen that the differentiation based on the Softsign linear–nonlinear tracking differentiator is better at the same frequency. Table 1 shows the root mean square values of the errors generated by the command filter and the Softsign linear–nonlinear tracking differentiator when the differentiation is extracted at different input frequencies, and it

can be seen that as the frequency increases, the differentiation of the command filter becomes less and less effective compared to the differentiation of the Softsign linear–nonlinear tracking differentiator.

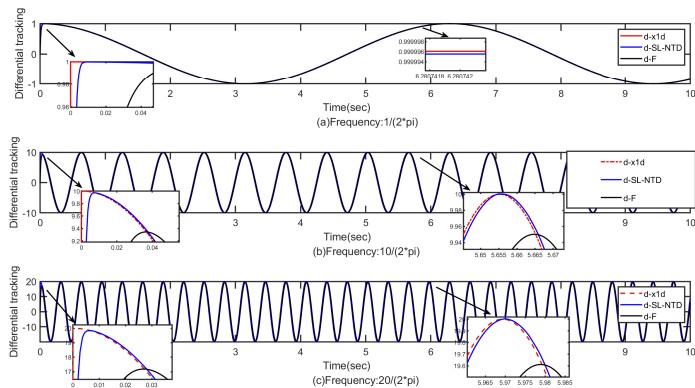

**Figure 8.** Differential trajectory of the Softsign linear–nonlinear tracking differentiator (d_SL-NTD) and the command filter (d_CF) at different input signal frequencies.

**Table 1.** RMS values for CF and SL-NTD errors extracted from differentiation at different input frequencies.

| Different Methods | | SL-NTD | CF |
|---|---|---|---|
| | 1/(2*pi) | 0.00082 | 0.00705 |
| RMS Value | 10/(2*pi) | 0.08192 | 0.70152 |
| | 20/(2*pi) | 0.32698 | 2.76532 |

The input signal was differentiated twice by a Softsign linear–nonlinear tracking differentiator, a command filter, and a conventional differentiator, respectively. As shown in Figure 9, it can be seen that the conventional differentiator exhibits a large overshoot of $10^{29}$, while the Softsign linear–nonlinear tracking differentiator exhibits an overshoot of $10^1$. It can be seen that the Softsign linear–nonlinear tracking differentiator can greatly mitigate the "complexity explosion" problem.

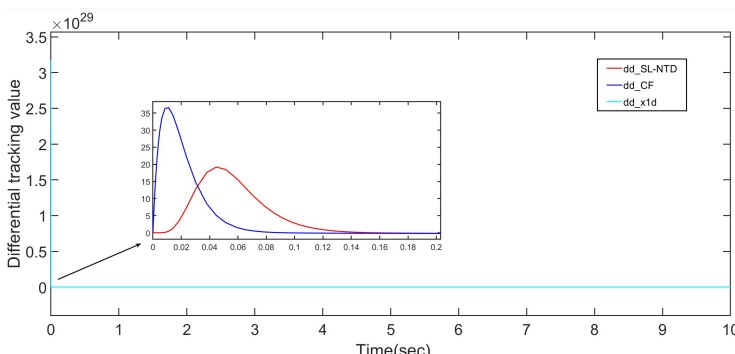

**Figure 9.** Tracking results of the SL-NTD, CF, and traditional differentiator after two differentiations.

When comparing the trajectory tracking of conventional backstepping control and command-filter-based backstepping control with the trajectory tracking of the Softsign excitation function-based tracking differentiator, as shown in Figure 10a, it can be seen that all three can achieve trajectory tracking, but the command-filter-based backstepping control and the conventional backstepping control fall behind at the beginning stage. As can be seen from Figure 10b, their trajectory tracking errors are more prominent, with the traditional backstepping control exhibiting the most significant errors, followed by

command-filter-based backstepping control, which has a larger overshoot at the beginning and takes longer to stabilize and still has periodic error fluctuations after stabilization. This shows that the trajectory tracking effect of the Softsign excitation function-based tracking differentiator for backstepping control proposed in this paper is significantly better than that of command-filter-based backstepping control and traditional backstepping control.

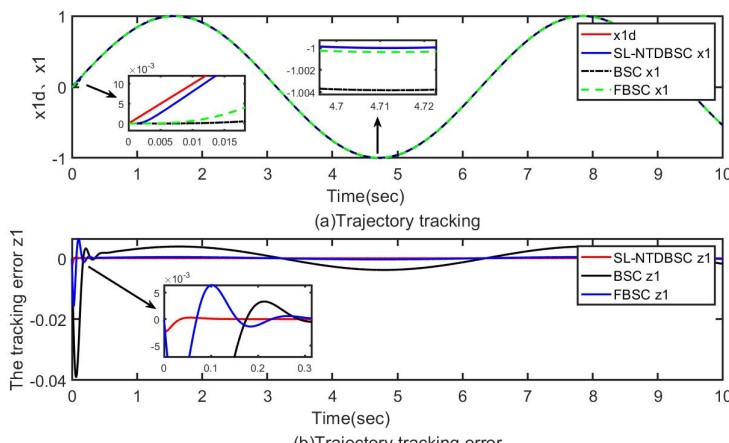

**Figure 10.** Comparison of trajectory tracking under FBSC and SL-NTDBSC.

To more strongly illustrate that the backstepping control of linear–nonlinear tracking differentiator based on the Softsign excitation function in this paper has a better trajectory tracking performance, it was compared with backstepping control using other tracking differentiators. The formulas for the tracking differentiators used for comparison are as follows:

Tracking differentiator 1: NTD [33]:

$$
\begin{cases}
\dot{x}_1(t) = x_2(t) \\
\dot{x}_2(t) = -R^2 * a * [x_1(t) - v(t) + tanh(x_1(t) - v(t))] \\
-R^2 * b * [x_2(t)/R + tanh(x_2(t)/R)]
\end{cases}
\tag{38}
$$

Tracking differentiator 2: LTD [27]:

$$
\begin{cases}
\dot{x}_1(t) = x_2(t) \\
\varepsilon^2 * \dot{x}_2(t) = -a_0 * (x_1(t) - v(t)) - a_1 * \varepsilon * x_2(t)
\end{cases}
\tag{39}
$$

Tracking differentiator 3: HSTD [28]:

$$
\begin{cases}
\dot{x}_1(t) = x_2(t) \\
\varepsilon^2 * \dot{x}_2(t) = -a_0 * (x_1(t) - v(t)) \\
-a_1 * |x_1(t) - v(t)|^m * sgn(x_1(t) - v(t)) - b_0 * \varepsilon \\
x_2(t) - b_1 * |\varepsilon * x_2(t)|^n * sgn(\varepsilon * x_2(t))
\end{cases}
\tag{40}
$$

The values of the parameters associated with the linear–nonlinear tracking differentiator based on the Softsign excitation function introduced in this paper and each of the other comparative tracking differentiators are as shown in Table 2.

Comparing the information in Table 2, it can be seen that the number of parameters of the linear–nonlinear tracking differentiator based on the Softsign excitation function is smaller compared to other comparative tracking differentiators, which makes the overall parameter adjustment process of the backstepping control designed in this paper easier.

Figure 11a shows a comparison of the tracking results of the differential signal for the backstepping control of the linear–nonlinear tracking differentiator based on the Softsign excitation function (SL-NTDBSC) with those of the backstepping control based on the linear tracking differentiator (LTDBSC), the backstepping control based on the high-speed tracking

differentiator (HSTDBSC) and the backstepping control based on the new nonlinear–linear tracking differentiator (NTDBSC). Referring to Figure 11a,b, it is obvious that SL-NTDBSC achieves the smallest trajectory tracking error with no excessive jitter and periodic errors, minimizing the error in this range.

**Table 2.** Parameter values of different tracking differentiators.

| Different Tracking Differentiators | Parameter Values |
| --- | --- |
| SL-NTD | $a = 7, b = 5$ |
| NTD | $R = 150, a_0 = 7, a_1 = 5$ |
| LTD | $\varepsilon = 0.004, a_0 = 2, a_1 = 1$ |
| HSTD | $\varepsilon = 0.004, a_0 = 5, a_1 = 0.5,$ $b_0 = 2, b_1 = 0.5, m = n = 0.5$ |

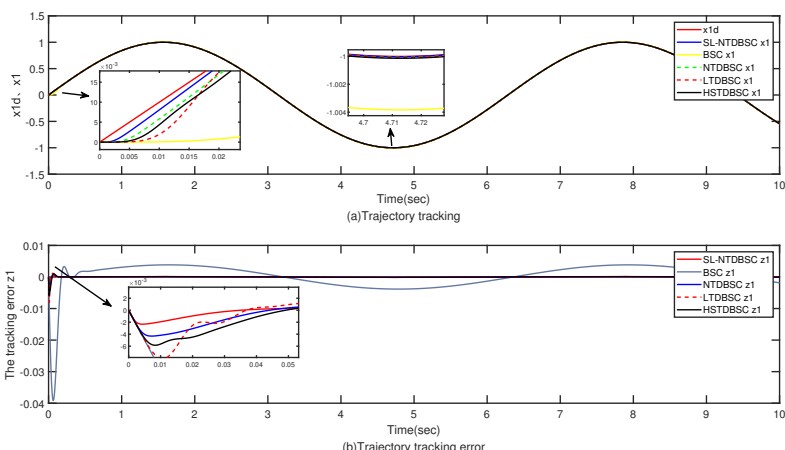

**Figure 11.** Comparison of trajectory tracking under backstepping control based on different tracking differentiators.

In order to quantitatively analyze the enhancement effect, two indicators were selected, as shown in Table 3 below.

$$I_{ITAE} = \int_0^t t|z_1(t)|dt \qquad (41)$$

$$I_{ISE} = \int_0^t [z_1(t)]^2 dt \qquad (42)$$

**Table 3.** Performance indexes of experimental data in Figure 11.

| Indicators / Controllers | ITAE Values | ISE Values |
| --- | --- | --- |
| SL-NTDBSC | 9.7414 | 0.00052 |
| NTDBSC | 17.3161 | 0.0016 |
| SL-NTDBSC lifting effect | 43.74% | 67.5% |
| LTDBSC | 20.7765 | 0.0025 |
| SL-NTDBSC lifting effect | 53.11% | 79.2% |
| HSTDBSC | 28.1929 | 0.0032 |
| SL-NTDBSC lifting effect | 65.45% | 83.75% |

The ITAE (integrated time absolute error) in Equation (41) is the absolute value of the error multiplied by the integration of the time term over time, which reflects both the magnitude of the error and the speed of error convergence, taking into account the control accuracy and the speed of convergence. The ISE in Equation (42) is the integral squared error, which indicates a large error in the suppression transition process. $z_1(i)$

are the position control errors at the i-th step of the control cycle. The control accuracy of SL-NTDBSC is the best and its convergence speed is the fastest, judging from the values of the $I_{ITAE}$ metric of SL-NTDBSC relative to other backstepping methods. In addition, the error in the transition process of SL-NTDBSC is almost zero, judging from the enhancement in the $I_{ISE}$ metric of SL-NTDBSC relative to other backstepping methods.

### 4.3. Experimental Results

To verify the results obtained from the simulations, experimental validation was carried out by using an accurate tracking platform and is detailed in this section. It is evident from the simulation analysis that the trajectory tracking performance of LTDBSC and HSTDBSC in backstepping control based on the improved tracking differentiator which was used for comparison is significantly weaker than that of NTDBSC. Therefore, the experiments in this paper compare the trajectory tracking performance of SL-NTDBSC, BSC, NTDBSC, and FBSC at an input signal amplitude of 100 and frequencies of 0.1, 0.5, 1.0, and 2.0.

As shown in Figure 12a, both conventional backstepping control and improved backstepping control can realize the tracking of the target trajectory when the frequency of the input signal is 0.1 Hz. However, the trajectory tracking performance of SL-NTDBSC and NTDBSC is better than that of the other methods, and the trajectory tracking errors of SL-NTDBSC and NTDBSC are minimized, which is also clearly seen in Figure 12b. When the frequency of the input signal is 0.5 Hz, as shown in Figure 13a, there is a decreasing trend in the trajectory accuracy of conventional backstepping control, and from Figure 13b, it can be seen that the amplitude of the trajectory tracking error of SL-NTDBSC is minimized. When the input signal frequency is 1.0 Hz, as shown in Figure 14a, the conventional backstepping controller cannot effectively control the trajectory accuracy, and overshooting occurs when the input waveform reaches the top or the bottom. It is clear from Figure 14b that the trajectory tracking error of conventional backstepping control is more than 20. The trajectory tracking errors of FBSC and NTDBSC are still larger than that of SL-NTDBSC. When the frequency of the input signal is 2.0 Hz, it can be seen from Figure 15 that conventional backstepping control is no longer able to achieve trajectory tracking. In Figure 15b, it can be seen that the active interval of the trajectory tracking error of FBSC and NTDBSC is almost two times that of SL-NTDBSC.

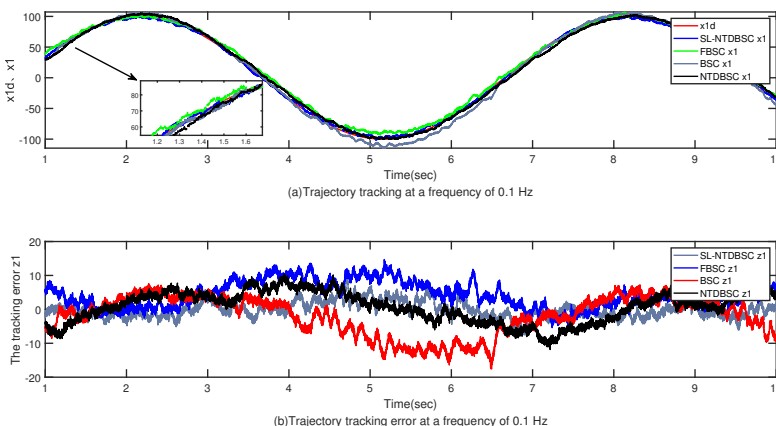

**Figure 12.** Trajectory tracking of different types of backstepping control at a frequency of 0.1 Hz based on an experimental platform.

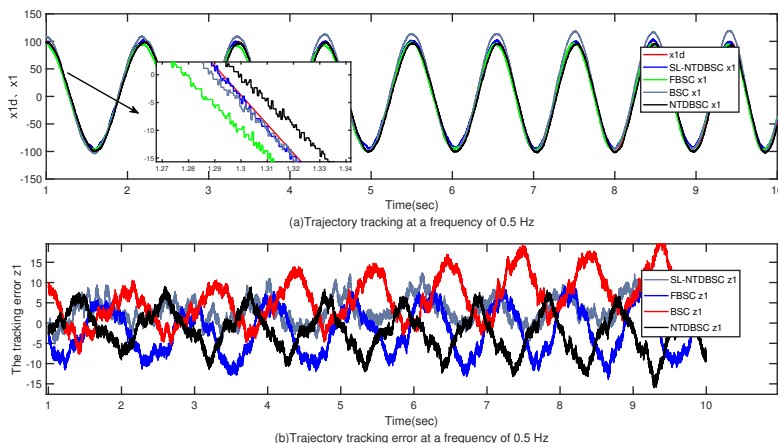

**Figure 13.** Trajectory tracking of different types of backstepping control at a frequency of 0.5 Hz based on an experimental platform.

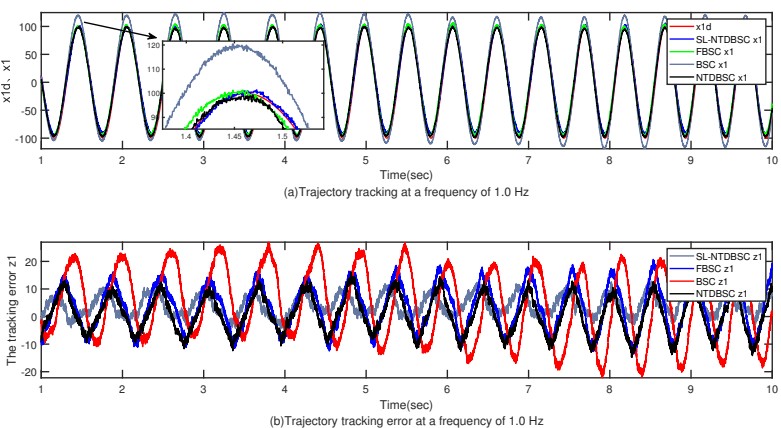

**Figure 14.** Trajectory tracking of different types of backstepping control at a frequency of 1.0 Hz based on an experimental platform.

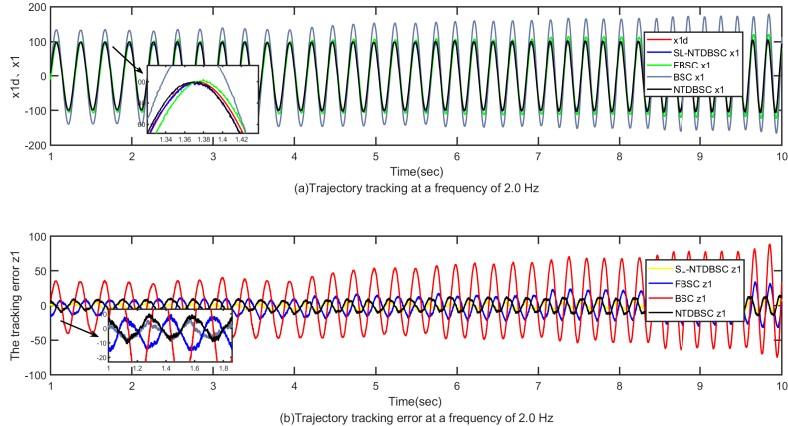

**Figure 15.** Trajectory tracking of different types of backstepping control at a frequency of 2.0 Hz based on an experimental platform.

In addition to the two quantitative analysis indexes used in the simulation, two new indexes were added to the experiment. All of these indexes can be used to measure the performance of a trajectory tracking control system from slightly different perspectives.

$$I_{MSE} = \frac{1}{N} \sum_{i=1}^{N} [z_1(i)]^2 \tag{43}$$

$$I_{APE} = \max_{i=1,\dots,N} |z_1(i)| \tag{44}$$

where $I_{MSE}$ denotes the mean square error index. It measures the average error of the system concerning the reference trajectory. The smaller the $I_{MSE}$, the smaller the average error of the system and the better the tracking performance. $I_{APE}$ is the absolute peak error index. The role of trajectory tracking is mainly to measure the difference between the pulse width of the control system output and the pulse width of the reference trajectory, which can reflect the accuracy of the control system. If the value of the $I_{APE}$ index is small, it means that the control accuracy of the system is high, and accordingly, the trajectory tracking performance is better. If the value of the $I_{APE}$ index is larger, it means that the control accuracy of the system is lower, and accordingly, the trajectory tracking performance may be poorer. $z_1(i)$ are the position control errors at the $i$-th step control cycle, and $N$ denotes the number of tracking error data for calculation.

Table 4 below shows the performance indexes for the data shown in Figures 12–15. Firstly, the $I_{ITAE}$ index was analyzed, from 0.1 Hz to 2.0 Hz. The values for SL-NTDBSC were smaller than those of the other backstepping methods used for comparison, which indicates that the system under SL-NTDBSC has the lowest overall degree of deviation from the reference trajectory and has the best tracking performance. At 2.0 Hz, the performance of SL-NTDBSC is improved by 70.92%, 90.19%, and 48.29%, respectively, compared to that of FBSC, BSC and NTDBSC. Secondly, when analyzing the $I_{ISE}$ index, from 0.1 Hz to 2.0 Hz, the value for SL-NTDBSC is consistently smaller than that of the other backstepping methods used for comparison, which indicates that the error is minimized over the whole time range under SL-NTDBSC and the system has the best tracking performance. Then, the $I_{MSE}$ index was analyzed, from 0.1 Hz to 2.0 Hz. The values for SL-NTDBSC are consistently smaller than those of other backstepping control methods, which indicates that the system has the smallest average error for the reference trajectory and the best trajectory tracking performance under SL-NTDBSC. Finally, after analyzing the $I_{APE}$ index from 0.1 Hz to 2.0 Hz, it is shown that the values for Sl-NTDBSC are still the lowest. The maximum tracking error of SL-NTDBSC is 12.8234 at 2.0 Hz, while the maximum tracking errors of the other backstepping methods are 34.7270, 89.0730, and 15.7490, respectively. Compared with BSC, the maximum tracking error of SL-NTDBSC is reduced by 85.60%, which indicates that the system has the highest control accuracy and the best trajectory tracking performance under SL-NTDBSC.

Comprehensively analyzing Figures 12–15 and Table 4, when the frequency of the input signal gradually increases, the trajectory tracking error of SL-NTDBSC changes less compared to other backstepping methods, where the change in the trajectory tracking error of BSC is especially large. The error that occurs at 2.0 Hz is larger, which shows that trajectory tracking was not realized.

When the input signal is a step signal, as shown in Figure 16, SL-NTDBSC has a faster response and a higher tracking accuracy when input signals are rapidly changed. However, in practice, due to the presence of noise and other influencing factors, these control methods will have jitter during tracking.

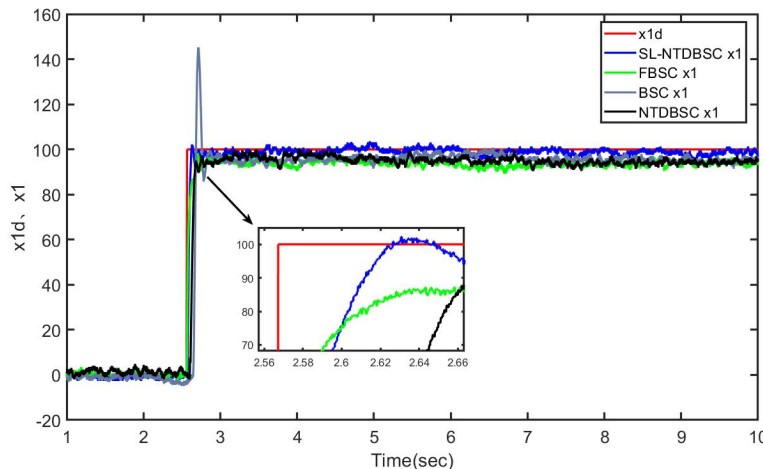

**Figure 16.** Trajectory tracking of various types of backstepping control based on the experimental platform when the input is a step signal.

**Table 4.** Performance indexes of experimental data in Figures 12–15.

| | $I_{APE}$ | | | |
| --- | --- | --- | --- | --- |
| | SL-NTDBSC | FBSC | BSC | NTDBSC |
| 0.1 Hz | 10.9908 | 14.7040 | 17.5949 | 11.8499 |
| 0.5 Hz | 12.3875 | 13.9040 | 21.1608 | 15.9570 |
| 1.0 Hz | 14.0081 | 20.7113 | 26.9787 | 16.0001 |
| 2.0 Hz | 12.8234 | 34.7270 | 89.0730 | 15.7490 |
| | $I_{ITAE}$ | | | |
| | SL-NTDBSC | FBSC | BSC | NTDBSC |
| 0.1 Hz | 7.6038 | 18.2346 | 19.6941 | 15.5492 |
| 0.5 Hz | 15.1563 | 22.5381 | 30.6350 | 18.7924 |
| 1.0 Hz | 19.0437 | 32.7671 | 49.3542 | 24.7648 |
| 2.0 Hz | 14.8539 | 51.0840 | 151.4505 | 28.7252 |
| | $I_{MSE}$ | | | |
| | SL-NTDBSC | FBSC | BSC | NTDBSC |
| 0.1 Hz | 5.5064 | 32.5650 | 35.4257 | 20.3340 |
| 0.5 Hz | 19.3630 | 36.8641 | 66.5298 | 27.5336 |
| 1.0 Hz | 30.7469 | 79.3472 | 184.1473 | 45.6778 |
| 2.0 Hz | 18.0900 | 162.8390 | 1443.10 | 52.0943 |
| | $I_{ISE}$ | | | |
| | SL-NTDBSC | FBSC | BSC | NTDBSC |
| 0.1 Hz | 4.1425 | 24.4990 | 26.6511 | 15.2975 |
| 0.5 Hz | 14.5670 | 27.7332 | 50.0511 | 20.7138 |
| 1.0 Hz | 23.1312 | 59.6937 | 138.5359 | 34.3638 |
| 2.0 Hz | 13.6093 | 122.5054 | 1085.70 | 39.1911 |

## 5. Discussion

The proposed backstepping control method in this paper based on a Softsign linear–nonlinear tracking differentiator addresses the "complexity explosion" issue present in traditional backstepping control. It reduces the number of parameters, simplifies parameter adjustment, and simultaneously balances signal convergence speed and stability, further enhancing the trajectory tracking performance of backstepping control. As electro-optical

tracking systems have increasingly diverse applications, with a variety of disturbance types and frequencies emerging, there is a need for further improvements to backstepping control to enhance the tracking and disturbance rejection capabilities. From the perspective of backstepping control design theory, fundamental nonlinear disturbances and uncertainties are treated as uncertain terms for suppression. Faced with diverse disturbance problems, one approach is to consider each disturbance as a new uncertain term and introduce a new disturbance estimation error. This can be achieved by using Lyapunov theory to estimate these uncertain terms, aiming to improve the disturbance rejection and tracking accuracy.

Furthermore, superior parameter adjustment methods will further enhance the tracking performance of optical tracking systems. Therefore, future work will involve designing a parameter adjustment method based on improved fuzzy control, combining it with improved backstepping control to simultaneously enhance the tracking precision and disturbance rejection capabilities in electro-optical tracking systems. This holds significant application value for electro-optical tracking systems.

## 6. Conclusions

To address the problems caused by electro-optical tracking systems with uncertainties, this paper employs backstepping control. However, traditional backstepping control is susceptible to the "complexity explosion" issue. Therefore, this paper introduces, for the first time, a backstepping control design based on a Softsign linear–nonlinear tracking differentiator. A novel tracking differentiator is designed using the Softsign function, enhancing the differentiation effect and providing some filtering capabilities. Simulation and experimental results confirm the effectiveness and superiority of this control method. The approach proposed in this paper overcomes the "complexity explosion" issue associated with traditional backstepping control by using a Softsign linear–nonlinear tracking differentiator to approximate the traditional differentiation process. After two consecutive differentiations, the overshoot peak is only $1/10^{28}$ of that of traditional differentiation, significantly reducing the likelihood of a complexity explosion. Moreover, this method reduces the number of parameters, simplifies parameter adjustment, and simultaneously balances signal convergence speed and stability, thereby improving the trajectory tracking performance of backstepping control. According to the ITAE index, over the frequency range of 0.1 Hz to 2.0 Hz, the tracking performance of this approach is improved by 65.88% compared to traditional backstepping control, 50.96% compared to command-filter-based backstepping control, and 35.46% compared to NTD-based backstepping control. Additionally, stability design using Lyapunov theory ensures the stability of this method, and it also guarantees the boundedness of system signals.

However, as the application scenarios of electro-optical tracking systems evolve, there will be an increased demand for control methods to have improved tracking precision characteristics and disturbance rejection capabilities. To meet the practical needs of electro-optical tracking systems, the next phase of research will focus on optimizing the method presented in this paper to further address the issue of the reduced disturbance rejection performance in multiple scenarios. Furthermore, forthcoming work will also consider the parameter adjustment problem in backstepping control, aiming to make parameter tuning more accurate and straightforward. In summary, the method proposed in this paper holds significant value for electro-optical tracking systems, and future research efforts will strive to enhance its performance and applicability in various real-world scenarios.

**Author Contributions:** Theoretical analysis: J.L. and S.Z.; Designing experiments and analyzing data: J.L. and H.W.; Conducting simulations: J.L. and S.Z.; Writing the paper: J.L.; Revising the paper: J.D. and Y.M. All authors have read and agreed to the published version of the manuscript.

**Funding:** This work was supported by the Special Research Assistant Program, Chinese Academy of Sciences, China (Grant No. E329691C21) and Natural Science Foundation of Sichuan Province for Youths, China (Grant No. 24NSFSC3777).

**Institutional Review Board Statement:** Not applicable.

**Informed Consent Statement:** Not applicable.

**Data Availability Statement:** Data are contained within the article.

**Conflicts of Interest:** The authors declare no conflicts of interest.

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
