# Peer review of "Design of Backstepping Control Based on a Softsign Linear–Nonlinear Tracking Differentiator for an Electro-Optical Tracking System"

_photonics, doi:10.3390/photonics11020156_

Round 1
Reviewer 1 Report
Comments and Suggestions for Authors
An application of the Softsign function on Backstepping Control is presented in this paper for an electro-optical tracking system. The paper presents mathematical models and theoretical frameworks for their proposed work. Overall the paper is well-written with experimental validations. However, it is not apparent that the significance of the research does not compare their method to some of the traditional methods.
Figure 12 (a), and Figure 15, need to be modified as they hide the time axis title.
Figure 11 (a), there is no need for multiple magnifications.
Comments on the Quality of English LanguageThe written quality is satisfactory.
Reviewer 2 Report
Comments and Suggestions for Authors
The article describes an important scientific issue.
The topic of the article is relevant.
I recommend accept this document after minor revision.
My decision is supported by the following comments:
1. Fig. 5 shows a photograph of the constructed test stand. Including a diagram of the test stand would be more legible.
2. The job description is very general. No parameters of the elements used are provided.
3. CCD is not the same as a camera. The detection matrix is one of the camera elements.
4. What camera was used (CCD or maybe CMOS)?
5. The Conclusion chapter is too short and too general and requires expansion.
Editorial page needs corrections:
- the legend in Fig. 1 should be in English
- no axis markings on the charts (fig. 1, 2)
Round 2
Reviewer 1 Report
Comments and Suggestions for Authors
The authors have addressed the issues/concerns that I raised, and have added relevant literature to improve the availability of references and also updated figures that needed to be updated.
The article can be accepted